

# The role of transcriptional and epigenetic modifications in astrogliogenesis

Shuangping Ma[1,*], Lei Wang[1,*], Junhe Zhang[1], Lujing Geng[2] and Junzheng Yang[1,3]

[1] Institutes of Health Central Plains, Tissue Engineering and Regenerative Clinical Medicine Center, Xinxiang Medical University, Xinxiang, China
[2] College of Life Sciences and Technology, Xinxiang Medical University, Xinxiang, China
[3] Guangdong Nephrotic Drug Engineering Technology Research Center, The R&D Center of Drug for Renal Diseases, Consun Pharmaceutical Group, Guangzhou, China
[*] These authors contributed equally to this work.

## ABSTRACT

Astrocytes are widely distributed and play a critical role in the central nervous system (CNS) of the human brain. During the development of CNS, astrocytes provide essential nutritional and supportive functions for neural cells and are involved in their metabolism and pathological processes. Despite the numerous studies that have reported on the regulation of astrogliogenesis at the transcriptional and epigenetic levels, there is a paucity of literature that provides a comprehensive summary of the key factors influencing this process. In this review, we analyzed the impact of transcription factors (*e.g.*, NFI, JAK/STAT, BMP, and Ngn2), DNA methylation, histone acetylation, and noncoding RNA on astrocyte behavior and the regulation of astrogliogenesis, hope it enhances our comprehension of the mechanisms underlying astrogliogenesis and offers a theoretical foundation for the treatment of patients with neurological diseases.

# INTRODUCTION

Neuroscience is an active research field with a multitude of fascinating yet unresolved issues. Epigenetics, a discipline that has experienced a period of rapid evolution in recent years, with a focus on phenomena that deviate from the principles of classical Mendelian genetics. This encompasses DNA modifications, RNA interference, and histone modifications, all of which play crucial roles and have propelled substantial advancements in neuroscience, particularly in the investigation of astrogliogenesis. These epigenetic mechanisms exhibit tremendous potential for elucidating the underlying processes of astrogliogenesis.

Astrocytes represent the most widely distributed and largest type of glial cells in the mammalian brain. While neurons are the primary cells responsible for processing information in the brain, astrocytes play a crucial supportive role in this process, for example, by aiding in neuronal migration and sustaining neuronal growth and development. Astrocytes are indispensable for the maturation of the central nervous system (CNS), the maintenance of the blood–brain barrier, the formation of synapses, and the transmission of neurotransmitters (*Bayraktar et al., 2014*; *Holst et al., 2019*). During the development of the infant brain, both neurons and astrocytes predominantly originate from

Corresponding author
Junzheng Yang,
yangjunzheng606403@163.com

neural stem cells. Astrocytes possess numerous long, branching projections that extend outward, giving them a star-like appearance. These projections are capable of establishing contact with neurons, blood vessels, and other glial cells, thereby forming a supportive network throughout the brain and spinal cord (*Volterra & Meldolesi, 2005*). It is noteworthy that astrocytes in the adult mouse brain display remarkable cellular plasticity, which enables them to be induced into neural progenitor cells that can subsequently generate neurons, astrocytes, and oligodendrocytes (*Zhang et al., 2022*). The activation of JAK/STAT signaling pathway, which is induced by ciliary neurotrophic factor (CTNF), leukemia inhibitors (LIF), and cytokine family members, has been demonstrated to promote astrogliogenesis (*Bonni et al., 1997*; *Jain et al., 2021*; *Kaur et al., 2005*). Furthermore, the Notch signaling pathway interacts with STAT signaling, particularly during organ development (*Hu et al., 2021*). The effector molecules of Notch signaling, Hes1 and Hes5, interact with STAT and JAK to promote STAT phosphorylation, thereby accelerating astrogliogenesis (*Yoshimatsu et al., 2006*). The evidence is increasingly suggestive that bone morphogenetic protein (BMP) signaling plays a role in axon regeneration, neuronal differentiation, synaptic maturation, and astrogliosis by facilitating cytoskeletal assembly, particularly in the context of CNS injury (*Akram et al., 2022*; *Zhong & Zou, 2014*). *Jiang et al. (2020)* demonstrated that exposure to nonylphenol, a stable environmental contaminant, during the perinatal period resulted in an increase in the number of astrocytes and a decrease in oligodendrogenesis in the offspring cerebellum, primarily by activating BMP signaling. The potential for astrocytes to be converted into neurons has been a topic of debate in recent years. Chen's laboratory has demonstrated that the gene NeuroD1, when delivered using an adeno-associated virus, can efficiently reprogram astrocytes in the grey matter into neurons (*Chen et al., 2020*; *Liu et al., 2020*). However, *Wang et al. (2021)* demonstrated that astrocyte-restricted NeuroD1 is unable to induce astrocyte-to-neuronal cell conversion whether in normal conditions or even after brain injury, using lineage-tracing strategies *in vivo*.

In addition to transcription factors, DNA epigenetic modifications have been demonstrated to play a crucial role in astrogliogenesis (*Albert & Huttner, 2018*; *Yoon et al., 2018*). A study demonstrated that astrocytes derived from the cerebellum and cortex exhibit a fundamental transcriptional and epigenomic program, yet display distinct cellular characteristics. This was revealed through an integrative analysis that included mRNA sequencing (mRNA-Seq), genome-wide DNA methylation sequencing, and Assay for Transposase-Accessible Chromatin using sequencing (ATAC-Seq) (*Welle et al., 2021*). The DNA methylation and demethylation at CpG dinucleotide sites exert a profound determination of neural stem cell fate. The DNA methyltransferases Dnmt1 and Dnmt3a have been demonstrated to impede the maturation of astrocyte differentiation by methylating astrocyte-specific genes, including GFAP and S100$\beta$ (*He et al., 2020*; *Urayama et al., 2013*).

A review of the role of multiple transcription factors and epigenetic modifications in astrogliogenesis provides a comprehensive perspective on the complex process, facilitating the elucidation of the molecular basis of astrocyte differentiation, the development of novel

new therapies for neurodegenerative diseases, and the promotion of the application of astrocytes in regenerative medicine.

## Survey methodology

This review comprises a synthesis of the research articles from the past five years in PubMed database, Scopus database, Embase database, ScienceDirect database, and Web of Science database.

# MECHANISM OF ASTROGLIOGENESIS AT TRANSCRIPTIONAL LEVEL

Transcription factors are essential molecules in the regulation of gene expression in eukaryotes. During the development of CNS, neural progenitor cells differentiate into either neuronal cells or glial cells with the differentiation process being guided by transcription factors.

## The role of NFI genes in astrogliogenesis

The nuclear factor I (NFI) is a family of transcription factors comprises four members, the four members of the family are NFIA, NFIB, NFIC, and NFIX. These factors regulate glial cell differentiation by controlling the expression of astrocyte markers in the developing CNS (*Brun et al., 2018*). NFIA, comprising 509 amino acids, has been shown to bind to the DNA sequence 5′-TGGCANNNTGCCA-3′ (*Gronostajski, 2000*). NFIA expression was enriched in astrocytes and oligodendroglia (*Chen et al., 2017*). Nfia-deficient mice exhibited abnormal glial cell development and corpus callosum abnormalities (*Shu et al., 2003*). NFIA expression was induced in the ventricular zone region at embryonic day 11.5 (E11.5) to maintain the pool of Glast-positive glial progenitor cells (*Deneen et al., 2006*). It facilitated the expression of the astrocyte marker GFAP by directly binding to its promoter (*Cebolla & Vallejo, 2006*), thereby supporting the notion that NFIA promotes the onset of early astrogliogenesis (*Tiwari et al., 2018*). Moreover, NFIA and SOX9 complex has been demonstrated to play a pivotal role in the initiation of astrogliogenesis (*Kang et al., 2012*). The aforementioned studies indicate that NFIA is a crucial and indispensable transcription factor during the process of astroglial progenitors differentiating into astrocytes. NFIB is a transcription factor predominantly localized in astrocytes and neuronal cells (*Chen et al., 2017*). It has been demonstrated that overexpression of NFIB in human pluripotent stem cells results in the formation of an astrocyte population within a period of two weeks (*Canals et al., 2018*; *Yeon et al., 2021*). Furthermore, NFIB-induced astrocytes exhibit physiological functions comparable to those of native astrocytes *in vivo*, as evidenced by RNA sequencing and cell function analysis (*Canals et al., 2018*). *Huang et al. (2022)* elucidated that the introduction of astrocyte-related transcription factors NFIB and SOX9 into chimeric human cerebral organoids (chCOs) accelerates the differentiation rate of induced pluripotent stem cells (iPSCs) into astrocytes. It is noteworthy that NFIB and SOX9 not only accelerate the differentiation of iPSCs into astrocytes, but their combination also enables the reprogramming of fibroblasts into astrocytes (*Quist et al., 2022*). It can be reasonably deduced that reprogrammed astrocytes, have the great potential to play

a pivotal role in maintaining brain homeostasis and addressing various neurological disorders. Furthermore, a study demonstrated that NFIX deletion in mice resulted in delayed development of both neuronal and glial lineages within the cerebellum (*Fraser et al., 2017*). Neurogenesis within the spinal cord remains normal in NFIX-deficient mice, aspects of terminal astrocytic differentiation are impaired (*Matuzelski et al., 2017*). It has been suggested that NFIX regulated the downstream of NFIA and NFIB coordinated gliogenesis within the spinal cord (*Matuzelski et al., 2017*).

## The role of JAK/STAT signaling in astrogliogenesis

Leukemia inhibitory factor (LIF) and its receptor (LIFR) have been demonstrated to play a protective role for cells of central nervous system, including neuronal cells, myelin oligodendrocytes, and astrocytes against apoptosis induced by hypoxic-glucose deprivation (*Huo, Fan & Wang, 2019*). Cytokines such as LIF or CNTF induce the dimerization of LIFR with the co-receptor gp130, leading to the phosphorylation and activation of Janus kinases (JAKs). This signaling pathway is of critical importance for the fate determination of glial lineages during brain development (*Bonni et al., 1997*). In the hippocampus of LIF-knockout mice, the number of astrocytes expressing GFAP was markedly decreased compared to wild-type mice (*Bugga et al., 1998*). Additionally, neural precursors in the developing forebrain of mice with reduced expression of LIFR were unable to generate astrocytes expressing GFAP and exhibit blocked neural differentiation (*Koblar et al., 1998*). Moreover, it was observed that gp130-deficient mice exhibited impaired differentiation of astrocytes (*Nakashima et al., 1999a*). It was observed that GP130, p-JAK2, and p-STAT3 were downregulated following calycosin treatment in $H_2O_2$-induced oxidative injury of spinal astrocytes (*Song et al., 2022*). This suggests that the repression of the JAK/STAT pathway contributes to the survival of spinal astrocytes. Neurogenin-1 (Ngn1) prevents the differentiation of neural stem cells and progenitor cells (NSCs) into astrocytes by inhibiting the JAK/STAT pathway (*Zhao et al., 2015*). Additionally, another molecule, neurogenin-2 (Ngn2), has been demonstrated to directly bind to the promoters of several astrocyte-specific genes and suppress their expression independently of STAT activity (*Sun et al., 2019*). A deficiency in zinc (Zn) has been demonstrated to impede astrogliogenesis during the prenatal period and in the context of developmental processes. This is achieved by affecting the activation of STAT3 signaling *via* a mechanism of redox regulation (*Supasai et al., 2021*). The transplantation of NSCs mainly predominantly results in the generation of astrocytes in the injured spinal cord (*Barnabe-Heider et al., 2010*; *Liu et al., 2015*; *Sabelstrom et al., 2013*). Spinal cord injury (SCI) is typically accompanied by the formation of scar tissue, and astrocytes derived from NSCs play a crucial role in limiting the expansion of the scar and preventing further axonal loss. In 2010, *Barnabe-Heider et al. (2010)* employed genetic fate mapping to ascertain provenance of nascent cells following an adult mouse SCI. The researchers observed that astrocytes and ependymal cells, which are typically restricted in their proliferation in the intact spinal cord, were the primary cell types undergoing proliferation following SCI (*Barnabe-Heider et al., 2010*). Another noteworthy study demonstrated the presence of Dil-labelled GFAP-expressing cells derived from NSCs in the post-injury period. Moreover, the differentiation of astrocytes derived

from NSCs was significantly higher than that of neuronal cells derived from NSCs in the spinal cord (*Liu et al., 2015*). *Zhu et al. (2011)* isolated NSCs from the subventricular region of newborn rats and observed that the addition of lithium to the medium resulted in a reduction in the number of astrocytes and inhibited the proliferation of glia-restricted progenitor cells (GRPs). Furthermore, the research demonstrated that lithium inhibited astrogliogenesis by inhibiting STAT3 activation *via* GSK-3$\beta$ (*Zhu et al., 2011*). JAK/STAT signaling, typically activated by cytokines, has been demonstrated to promote astrocyte production, while forced activation of JAK/STAT signaling has been observed to result in precocious astrocyte formation. Conversely, the inhibition of this pathway has been demonstrated to prevent the astrocyte differentiation (*He et al., 2005*). *Wang et al. (2020a)* clarified that UCA1 expression was significantly inhibited in rats with temporal lobe epilepsy (TLE) induced by kainic acid (KA). The overexpression of UCA1 in these rats resulted in a prolonged latency period in the Morris water maze assay and a reduction in the number of GFAP-expressing cells in the hippocampus. The co-localization of GFAP-positive cells and p-STAT3-positive cells confirmed that STAT3 activation occurred in astrocytes. UCA1 decreased the expressions of p-JAK1 and p-STAT3, indicating that KA-induced astrocyte activation is inhibited of *via* the JAK/STAT signaling pathway (*Wang et al., 2020a*). Moreover, this inhibitory effect was mainly mediated by the JAK/STAT signaling pathway in the temporal lobe epilepsy (*Wang et al., 2020a*). Furthermore, the blockade of the JAK/STAT3 pathway in reactive astrocytes has been demonstrated to increase the frequency of Huntington's aggregates, a hallmark of Huntingtin's disease (*Ben Haim et al., 2015*). Given the specificity of the JAK/STAT pathway in regulating cell proliferation, neuroinflammation, and astrocyte differentiation, it has become a potential drug target for neurological diseases, such as Alzheimer's disease (*Desai et al., 2020*; *Jain et al., 2021*).

## The role of BMP signaling in astrogliogenesis

Bone morphogenetic proteins (BMPs) constitute a family of proteins within the transforming growth factor beta superfamily, and BMP-mediated signaling pathways play a pivotal role in the process of astrogliogenesis (*Huang & Xiong, 2016*). In the CNS, BMPs facilitate the generation of astrocytes while simultaneously inhibiting the differentiation of oligodendrocyte progenitor cells (OPCs) (*Costa et al., 2019*). During the early stage of development, BMPs are expressed in the lateral edge of the neural plate and subsequently in the dorsal midline of the neural tube, and this promote dorsal-medial patterning and the development of the ventral forebrain (*Gámez, Rodriguez-Carballo & Ventura, 2013*; *Mehler et al., 1997*). BMPs are widely expressed during late development, especially in the hippocampus and cerebral cortex (*Gámez, Rodriguez-Carballo & Ventura, 2013*; *Mehler et al., 1997*). Furthermore, YAP signaling is indispensable for the formation of neocortical astrocytes. *Huang & Xiong (2016)* elucidated that neogenin is necessary for the activation of RhoA by BMP2 and revealed that neogenin promotes the formation of neocortical astrocytes through the cascade reaction of BMP2-Neogenin-YAP-Smad1 cascade reaction (*Huang & Xiong, 2016*; *Wu et al., 2021*) (Fig. 1). In BMP4 transgenic mice, an increase in astrocyte density of 40% was observed in several brain regions, while oligodendrocyte

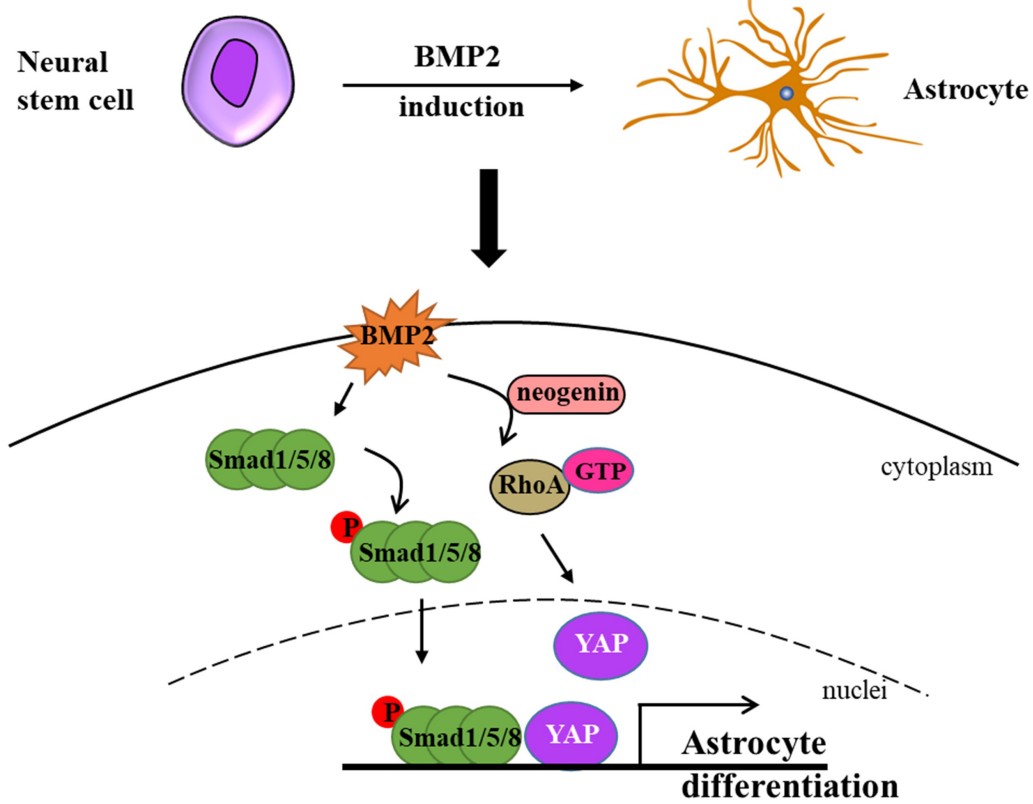

**Figure 1 Neural stem cells can differentiate into astrocytes under BMP2-induction.** Briefly, BMP2 promotes p-Smad1/5/8 getting into nuclei in the presence of YAP and activates RhoA assisted by neogenin. Ultimately, BMP2 accelerates astrocyte differentiation without the cooperation of p-Smad1/5/8 signaling and YAP signaling (modified from *Huang & Xiong, 2016*: Fig. 1).

density decreased by 11% to 26% (*Gomes, Mehler & Kessler, 2003*). In a recent study, *Lattke et al. (2021)* demonstrated that the addition of BMP4 to culture conditions promoted the differentiation of neural stem cells (NSCs) into astrocytes, which is consistent with a previous study in the field. Fibrinogen derived from blood has been demonstrated to significantly enhance astrogliogenesis *via* the activation of the BMP receptor signaling pathway while simultaneously inhibiting neuronal differentiation in the subventricular zone (SVZ) and hippocampal neural stem cells (NSCs) (*Pous et al., 2020*). Furthermore, the absence of fibrinogen markedly impairs astrocyte formation following cortical injury (*Pous et al., 2020*). In addition, simultaneous knockout of BMPR1a and BMPR1b in mice has been shown to reduce the number of mature glial cells in the neural tube by 25% to 40% (*See et al., 2007*). Introducing TGF-$\beta$1 *via* intraventricular in-utero injection resulted in the disorganization of radial glial fibers and premature gliogenesis with the appearance of GFAP-positive cells (*Stipursky et al., 2014*).

## Other transcription factors in astrogliogenesis

WNT/$\beta$-catenin, as a classic transcriptional regulator, plays a pivotal role in maintaining the equilibrium between cell proliferation and differentiation during neurogenesis.

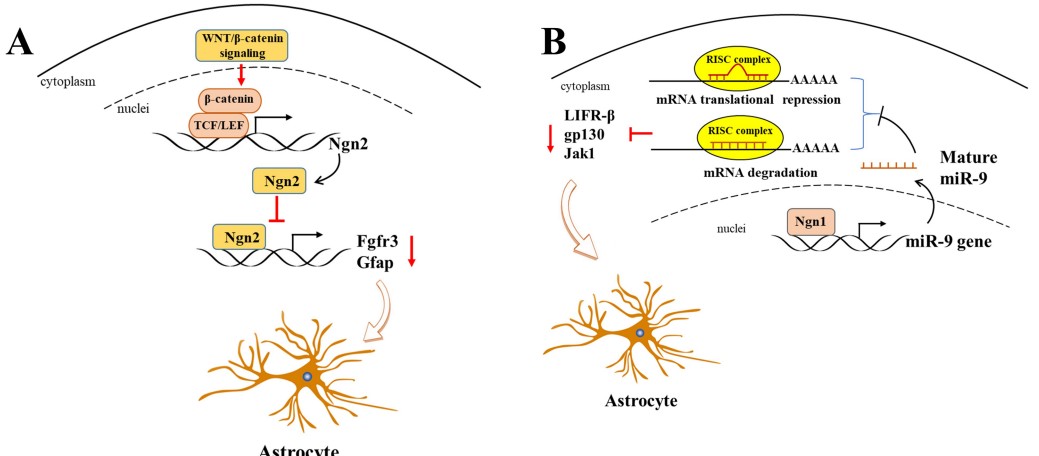

**Figure 2** **Ngn2 and Ngn1 participate in the astrogliogenesis.** (A) WNT/ $\beta$-catenin promotes Ngn2 expression by binding to the promoter of Ngn2 *via* putative TCF/LEF binding sites. Next, increased Ngn2 inhibits astrocytic gene Fgfr3 and Gfap transcription by binding to their promoters, leading to restriction of astrogliogenesis. (B) Ngn1 activates miR-9 expression *via* promoting its transcription, and mature miR-9 decreases the expression of JAK/STAT upstream molecules, such as LIFR- $\beta$, gp130, and Jak1, by binding to their 3′ UTR. ultimately leading to inhibition of astrogliogenesis (modified from *Huang & Xiong, 2016*: Fig. 1).

Furthermore, its activation results in an increased expression of neurogenin 2 (Ngn2), which in turn disrupts astrogliogenesis in the developing spinal cord (*Sun et al., 2019*). It is of particular significance that Ngn2, acting as a transcription repressor, directly binds to astrocytic gene promoters, including *Fgfr3* and *Gfap*, and thereby inhibits their transcription, thus impeding astrogliogenesis in the developing spinal cord (*Sun et al., 2019*) (Fig. 2A). *Gan et al. (2014)* demonstrated that the deletion of $\beta$-catenin in hGFAP-Cre mice inhibited neocortex formation, primarily by disrupting the development of radial glial cells. Furthermore, the JAK-STAT pathway, another classic transcription factor pathway, has also been demonstrated to play a role in astrogliogenesis. For example, *Zhao et al. (2015)* demonstrated that the molecule Ngn1 binds to the promoter of miR-9 to promote its expression in the brain. miR-9 directly targets key upstream molecules of the JAK-STAT signaling pathway, such as LIFR-$\beta$, Il6st (gp130), and Jak1, preventing STAT phosphorylation and ultimately leading to the inhibition of astrogliogenesis (Fig. 2B).

The transcription factor zinc finger and BTB domain-containing protein 20 (ZBTB20) is also involved in cell-specific regulation during neocortical development, particularly in the process of astrogliogenesis. The protein is expressed in the proliferative regions of the mouse neocortex during embryonic stages E15 to E18 and is also transiently expressed in projection neurons of the upper cortex during the early postnatal period. In adults, its expression is primarily limited to astrocytes (*Mitchelmore et al., 2002*; *Nagao et al., 2016*; *Tonchev et al., 2016*). Recently, the role of ZBTB20 in modulating astrogliogenesis has been a topic of contention in recent studies. It has been demonstrated that ZBTB20 is highly expressed in late-stage NPCs and their astrocytic progeny. While overexpression of ZBTB20 promoted astrocytogenesis and its knockdown suppressed it, this process was

dependent on Sox9 and NFIA, and not through direct activation of the *Gfap* promoters (*Nagao et al., 2016*). However, *Medeiros de Araújo et al. (2021)* demonstrated that in mice in which ZBTB20 was conditionally deleted, the number of astrocyte subsets expressing with SOX9 and GFAP were increased after P10, while the number of S100$\beta$+ cells remained unaltered. Furthermore, the expression of *Zbtb20*$^{WT}$ at E16.5 was observed to significantly increased the number of S100$\beta^+$ astrocytes, while the expression of *Zbtb20*$^{DN}$ (DN: dominant-negative mutation) which is associated with Primrose syndrome (*Cordeddu et al., 2014*), was found to significantly reduced their number of astrocytes. This was attributed to its impact on the collaboration with other ZBTB family members to regulate astrogliogenesis (*Medeiros de Araújo et al., 2021*). SOX9 is an important transcription factor that is astrocyte-specific and directly regulates GFAP expression by binding to its promoter (*Byun et al., 2020*). It facilitates astrogliogenesis by coordinating with NFIA (*Kang et al., 2012*) and targeting the JAK-STAT and BMP pathways.

## MECHANISM OF ASTROGLIOGENESIS AT THE EPIGENETIC LEVEL

### DNA methylation

DNA methylation represents a crucial epigenetic mechanism in mammalian genomes, whereby alterations in gene expression can be achieved without any modification to the underlying DNA sequences. In bacteria, DNA methylation provides the regulatory basis for host adaptation to various environments and serves as a defense method to protect bacterial DNA from foreign phage DNA (*Bhootra et al., 2023*; *Casadesús & Sánchez-Romero, 2022*). This process is catalyzed by a group of enzymes known as DNA methyltransferases (Dnmts), which transfer a methyl group from S-adenosine methionine (SAM) to the fifth carbon of a cytosine residue, resulting in the formation of 5-methylcytosine (5mC) (*Moore, Le & Fan, 2013*; *Nakagawa et al., 2020*). It should be noted that DNA methylation is not limited to gene promoters; it also occurs in gene bodies, enhancers, silencers, and transposons (*Wang & Xu, 2014*). When DNA methylation occurs at gene promoters, it has the potential to modulate gene expression by recruiting proteins involved in gene suppression or by affecting the binding of transcription factors to DNA (*Jaenisch & Bird, 2003*; *Moore, Le & Fan, 2013*).

DNA methylation is a key process in the differentiation of astrocytes during the fetal stage of brain development. At E11.5 days, the GFAP promoter, which is bound by STAT3, underwent rapid demethylation in response to STAT3 signaling activation (*Takizawa et al., 2001*). Over 90% of cells within CNS displayed DNA hypomethylation, which resulted in the enhanced expression of GFAP and S100$\beta$. This ultimately resulted in premature astrocyte differentiation in mice with conditional deletion of DNMT1 (*Fan et al., 2005*). However, the absence of DNA methylation in the GFAP promoter is not sufficient for STAT3 binding to its target sites. The knockout of DNMT1, DNMT3a, and DNMT3b in mouse embryonic stem cells resulted in the complete demethylation on the GFAP promoter region, however, STAT3 was unable to bind to the GFAP promoter to induce GFAP transcription, instead, it induced Socs3 transcription which is another target for the JAK-STAT signaling pathway

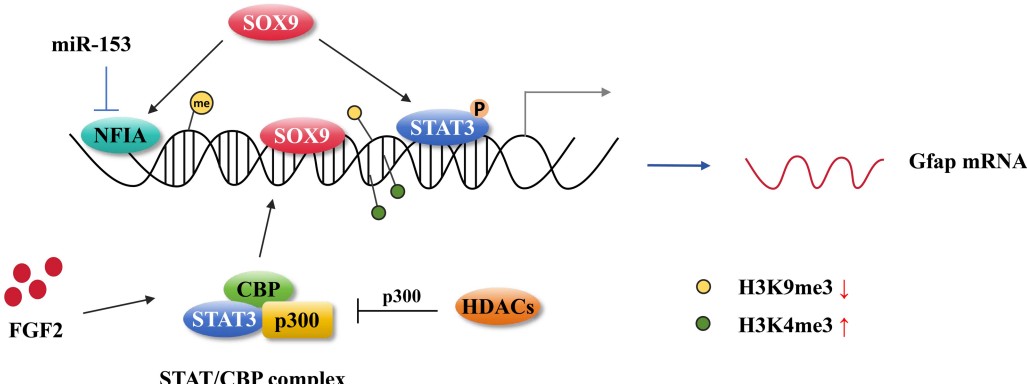

**Figure 3  Astrocytic gene regulation in astrocyte generation.** MiR-153 inhibits NFIA expression at the transcriptional level and impedes NFIA binds to the distal promoter of GFAP. The activation of STAT3 promotes the hypomethylation of the GFAP promoter. SOX9 directly binding to the GFAP promoter activates GFAP expression and it also can cooperate with NFIA or STAT3 affecting GFAP expression. FGF2 facilitates STAT/CBP complex accessing to GFAP promoter by increasing the level of H3K4me3 in STAT binding sites and inhibiting the level of H3K9me3. HDAC genes compete with the STAT/CBP complex on p300.

(*Urayama et al., 2013*) (Fig. 3), suggesting that hypomethylation in the GFAP promoter containing the STAT3-binding site alone is not sufficient to induce GFAP expression. It is of particular importance to note that the accessibility of STAT3 to the binding site in the GFAP promoter region is a crucial factor in the process of astrogliogenesis. The forced expression of NFIA in NSCs at E11 results in the dissociation of DNMT1 from the GFAP promoter, thereby preventing the late NSCs from replicating the methylation signature on the newly synthesized chain in the daughter cells (*MuhChyi et al., 2013*; *Namihira et al., 2009*). In mice with a conditional mutation in Dnmt1, there were approximately 90% of cortical and hippocampal cells from E13.5 in the dorsal forebrain exhibited hypomethylation (*Hutnick et al., 2009*).

## Histone modifications

In mammalian genomes, a nucleosome is composed of an octamer consisting of two molecules each of histones H2A, H2B, H3, and H4, with 147 bp DNA wrapped around it (*Chong & Gan, 2023*). While the core part of the histone in the nucleosome remains relatively uniform, the free N-terminal tails are subject to a range of modifications, including acetylation, methylation, phosphorylation, ubiquitination, and ADP ribosylation. These modifications play a crucial role in influencing the transcriptional activity of genes (*Zhang et al., 2021*).

The diverse range of histone modifications can either inhibit or activate gene transcription depending on whether they lead to chromatin closure or opening, respectively. This ultimately influences the expression of astrocyte lineage genes. Setdb1 (Eset), a histone H3K9-specific methyltransferase, has been demonstrated to inhibit the gene expression through its interaction with the co-repressor KAPI (*Nakagawa et al., 2020*). Setdb1 is highly expressed during the early stages of mouse brain development. As Setdb1

expression decreases over time, the level of H3K9 trimethylation also decreases, resulting in dysregulation of the specific neural cell gene expressions (*Tan et al., 2012*). However, Setdb1 has also been observed to activate the expression of certain non-neural genes, including those related to the astrocyte lineage, thereby promoting the formation of astrocytes (*Tan et al., 2012*). It has been reported that various chromatin marks engage in crosstalk at regulatory elements in gene expression. Specifically, during the stages of astrogliogenesis, ChIP-seq profiles revealed that the levels of H3K27ac and H3K4me1 are enriched at astroglial genes, including Gfap and Aqp4 (*Tiwari et al., 2018*). have been demonstrated to impede the process of HDAC genes by regulating the acetylation level of STAT3 and competing with STAT3 for binding to p300 (*Zhang et al., 2016*). In CKO-Olig1 mice, the loss of HDAC3 has been observed to interfere with neurogenesis and astrocyte development, leading to myelination defects and altered astrocyte responses (*Zhang et al., 2016*). The STAT-SMAD-p300/CBP complex is essential for the expression of astrocyte-specific gene. Furthermore, it has been demonstrated that STAT and BMP signaling act in a synergistic manner to regulate the astrocyte gene expression (*Nakashima et al., 1999b*). Histone H3K4me3 is also critical for gene expression regulation, particularly in transcriptional activation. Fibroblast growth factor 2 (FGF2) has been demonstrated to confer the potential for NSCs to differentiate into astrocytes by modifying histone H3 *in vitro* (*Song & Ghosh, 2004*). Although FGF2 itself does not induce GFAP expression, it promotes the binding of the STAT/CBP complex to the GFAP promoter by increasing the level of H3K4me3 at STAT binding sites and reducing the level of H3K9me3, thereby enhancing CNTF-induced astrogliogenesis (*Song & Ghosh, 2004*) (Fig. 3). It has been demonstrated that the MAPK signaling is activated by FGF in the process of cortical development, which is critical to the fate determination of NSCs differentiation into neural cells or astrocytes (*Dinh Duong et al., 2019*). *Hirabayashi & Gotoh (2010)* provided the genetic evidence to support the hypothesis that the histone modifications play a role in astrocyte formation. It has been demonstrated that the polycomb protein EZH2 inhibited the premature differentiation of glial cells in neural progenitor cells (*Sparmann et al., 2013*). This inhibition occurs as a result of EZH2 preventing GFAP expression by interacting with the chromatin helicase DNA-binding protein 4 (CHD4). In CHD4-deficient mice, astrogenesis was enhanced during neocortex development (*Sparmann et al., 2013*). Additionally, *Sher, Boddeke & Copray (2011)* reported that EZH2 expression in astrocytes induced dedifferentiation of astrocytes back into neural stem cells (NSCs). Moreover, another epigenetic modification, histone serotonylation, has been identified as a factor in astrocytic Slc22a3-regulated sensory processing (*Sardar et al., 2023*). The aforementioned studies suggest that the differentiation of neural stem cells (NSCs) into astrocytes, as well as the generation and functional roles of astrocytes, are subject to a number of epigenetic modification processes.

## Noncoding RNAs

MicroRNAs (miRNAs) which are approximately 21 nucleotides in length and bind to the $3'$ untranslated region (UTR) of target genes, they regulate growth, development, and various cellular processes by promoting the degradation or hindering the translation of

these target genes (*Krol, Loedige & Filipowicz, 2010*; *Lu & Rothenberg, 2018*). Mounting evidence has demonstrated that miRNAs participate in the regulation of astrogliogenesis at the post-transcriptional level. For example, overexpression of miR-153 in the developing cortex *via* lentiviral infection resulted in a decrease in astrocytes expressing the bubblegum family member 1 (ACSBG1), while increasing the number of neuronal cells expressing NeuN$^+$ (*Tsuyama et al., 2015*). The 3′ untranslated regions of NFIA and NFIB containing binding sites for miR-153, and they expressions were downregulated when miR-153 was overexpressed in sensory neurons and telencephalic neurons (*Tsuyama et al., 2015*). miR-153 targeted NeuroD1 in glial cells and was involved in LPS-induced neuroinflammation through inhibiting the phosphorylation of MAPK signaling (*Choi et al., 2022*).

miR-124 has been demonstrated to play a pivotal role in the reprogramming of astrocytes into immature neuronal cells. This process is initiated by direct targeting of the RNA-binding protein Zfp36L1 at the post-transcriptional level, which is crucial for the successful reprogramming of these cells. This interaction is associated with ARE-mediated mRNA decay, which subsequently inhibits the neurogenic interaction group of Zfp36L1 (*Papadimitriou et al., 2023*). In addition to post-transcription regulation, miR-124 contributes to neurogenesis by interacting with polypyrimidine-tract-binding protein (PTBP) to regulate nervous system-specific alternative splicing patterns (*Wang et al., 2023*). In the agomiR-124 treated SCI model, mRNA sequencing revealed a total of 85 upregulated genes and 80 downregulated genes, of these, Tal1 was identified as a potential target gene of miR-124 that mediated the proliferation and differentiation of neuronal precursor cells (*Wang et al., 2020b*). In another study, miR-124 overexpression was observed to promote the proliferation and neural differentiation in NSCs, while overexpression of its target gene DLL4 was found to reverse these promotive effects (*Jiao et al., 2017*). Furthermore, miR-124 has been demonstrated to induce the neuronal generation by acting on endogenous neurogenetic pathways, and the addition of the neurogenic compound ISX9 greatly promoted the differentiation and functional maturation of induced neurons (*Papadimitriou et al., 2023*). These results implicate that miR-124 facilitates neuronal differentiation by targeting Zfp36L1, PTBP1, Tal1, DLL4, and other mechanisms. The question of whether the RNA-binding protein PTB (also known as PTBP1) can induce astrocytes to differentiate into neurons has the subject of considerable debate in recent times. In 2020, *Qian et al. (2020)* reported that the depletion of PTBP1 *via* lentivirus in mouse cortical astrocytes resulted in the conversion of astrocytes into neurons, as evidenced by the expression of pan-neuronal markers Tuj1 and MAP2. Similarly, in the same year, Yang's laboratory demonstrated that the efficient generation of dopaminergic neurons from Müller glia could be achieved by knocking down PTBP1 using the CRISPR system CasRx (*Zhou et al., 2020*). However, subsequent studies presented evidence that challenges the hypothesis that PTBP1 can convert astrocytes into neuronal cells. Wang et al. confirmed that *in vivo* knockdown of PTBP1 did not result in the conversion of resident astrocytes into neurons, as evidenced by lineage tracing strategies (*Hoang et al., 2023*; *Wang et al., 2021*). Another two additional studies have also reported that the knockdown or depletion of PTBP1 was unable to convert astrocytes into hippocampal neurons or dopaminergic neurons in Alzheimer's mouse models or Parkinson's mouse models, respectively (*Chen*

*et al., 2022*; *Guo et al., 2022*). These contrasting results suggest that the consequences of PTBP1 knockout at the genomic level may differ from those of knockdown at the mRNA level. These findings underscore the significance of genetic manipulation, lineage tracing, and gene expression profiling in scientific research.

The brain-specific molecule miR-9/9* (*Jung et al., 2012*), which directs chromatin remodeling in conjunction with miR-124, targets BAF53a at the post-transcriptional level (*Yoo et al., 2009*). Overexpression of miR-9/9* and miR-124 in neonatal retinal progenitor cells by electroporation *in vivo* also affected the cell fate of glial cells, these cells were observed to have reduced the expressions of SOX9 and GS, and increased the level of TUBB3 in the postnatal day 14 retina (*Suzuki et al., 2020*). It suggests that miR-9/9* and miR-124 tend to induce the differentiation of NSCs and retinal progenitor cells into neuronal cells.

It has been reported that in the members of Let-7 family of miRNAs regulate the timing of glial differentiation. Let-7 plays a pivotal role in the determination of the fate of neural progenitor cells, mainly through the targeting of the chromatin-associated protein HMGA2 (*Patterson et al., 2014*), which promoted Notch intracellular domain entry into Hes5 promoter and subsequently induced astrogliogenesis (*Patterson et al., 2014*; *Rodríguez-Rivera et al., 2009*). Induced deletion of let-7 and miR-125 in glial progenitors inhibits astrocyte differentiation (*Shenoy, Danial & Blelloch, 2015*). Due to Dgcr8 encodes a key cofactor in miRNA generation, *Shenoy, Danial & Blelloch (2015)* demonstrated that BMP signaling was not affected in Dgcr8-knockout mice, but that pSTAT signal was significantly impaired astrocyte differentiation. The reexpression of let-7 and miR-125 in glial progenitor cells was observed to significantly rescue the differentiation disorders caused by Dgcr8 defects, and this rescued mechanism was found to be parallel to JAK-STAT signaling activating astrogliogenesis (*Shenoy, Danial & Blelloch, 2015*). In future studies, it would be interesting to determine how the fusion of the JAK-STAT pathway and miRNA to promote astrocyte formation.

## CONCLUSIONS AND PERSPECTIVE

In conclusion, this paper provides an overview of the key transcription factors, including NFIA, NFIB, and JAK/STAT signaling, which have been demonstrated to play pivotal roles in directing neural stem cells towards an astrocytic fate. NFIA and NFIB are particularly crucial for astrocyte differentiation, whereas JAK/STAT signaling is activated by cytokines such as LIF and CNTF, which facilitate astrogliogenesis. BMP signaling, represents another crucial pathway, facilitating astrogliogenesis while inhibiting oligodendrocyte differentiation. Additionally, other factors, including WNT/$\beta$-catenin, ZBTB20, and microRNAs, have been demonstrated to influence this process, either by promoting or inhibiting astrocyte development. At the epigenetic level, DNA methylation, histone modifications, and noncoding RNAs play crucial roles in regulating gene expression during astrogliogenesis. The action of DNA methyltransferases, such as DNMT1, has been demonstrated to inhibit the astrocyte differentiation by methylating astrocyte-specific gene promoters. Histone modifications, including acetylation, methylation, and ubiquitination,

affect chromatin accessibility and, as a consequence, regulate astroglial gene transcription. MicroRNAs, including miR-124, miR-9, and Let-7 family members, facilitate the process by targeting specific transcripts and modulating their expression post-transcriptionally. These molecular mechanisms not only enhance our comprehension of astrogliogenesis but also provide novel perspectives and potential therapeutic avenues for the treatment of neurological diseases. Furthermore, it is anticipated that this will facilitate the application of astrocytes in regenerative medicine. The presence of astrocytes is of significance to diseases that have communication barriers with synapses or axons. Perhaps, astrocytes have great inherently plastic with regard to their capacity to "listening to", "talking to", and form memories of neuronal cells. Future research could investigate the interactions between transcription factors and epigenetic modifications in greater depth, elucidating their specific mechanisms of action across a range of disease models. Moreover, the development of targeted therapeutic drugs against these specific points holds considerable promise for offering novel strategies and therapeutic approaches for neurological disorders, including Parkinson's disease (PD), Alzheimer's disease (AD), and epilepsy.

### Funding

This work was supported by the National Natural Science Foundation of China (82202049), the program for Science and Technology Innovation Team in Universities of Henan Province (22IRTSTHN030), and the Key Research and Development and Promotion Special Project of Henan (Scientific and Technological Project) (222102310517). The funders had no role in study design, data collection and analysis, decision to publish, or preparation of the manuscript.

### Grant Disclosures

The following grant information was disclosed by the authors:
National Natural Science Foundation of China: 82202049.
Universities of Henan Province: 22IRTSTHN030.
Key Research and Development and Promotion Special Project of Henan: 222102310517.

### Competing Interests

Junzheng Yang is employed by Consun Pharmaceutical Group.

### Author Contributions

- Shuangping Ma conceived and designed the experiments, prepared figures and/or tables, and approved the final draft.
- Lei Wang analyzed the data, prepared figures and/or tables, and approved the final draft.
- Junhe Zhang conceived and designed the experiments, performed the experiments, prepared figures and/or tables, authored or reviewed drafts of the article, and approved the final draft.
- Lujing Geng performed the experiments, authored or reviewed drafts of the article, and approved the final draft.

- Junzheng Yang conceived and designed the experiments, prepared figures and/or tables, authored or reviewed drafts of the article, and approved the final draft.

## Data Availability

This is a literature review.

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
