# Peer review of "The role of transcriptional and epigenetic modifications in astrogliogenesis"

_PeerJ, doi:10.7717/peerj.18151_

## Round 0.1 · original submission · Major Revisions

· Academic Editor

Major Revisions

Thank you for submitting your paper to PeerJ for consideration.

Your paper has now been assessed by four independent reviewers and I am pleased to say that, on the basis of the reports received from them, the Journal is more than willing to consider your manuscript, providing you and the co-authors make the revisions that have been suggested by the reviewers. There are a large number of revisions suggested, which I feel are helpful and fairly straightforward changes to make.

Reviewer 1 ·

Basic reporting

This review comprehensively summarizes the role of transcription factors and epigenetic modifications in astrogliogenesis. It is well-written and cites relatively recent publications. It offers valuable insights for researchers studying astrogliogenesis and neurological diseases.

Experimental design

no comment

Validity of the findings

no comment

·

Basic reporting

The authors in this article have compiled an extensively researched review article on the transcriptional and epigenetic control of astrogliogenesis. Their goal to summarize the regulation of astrogliogenesis is to provide potential prospects and strategies to alleviate and treat neurological diseases.

1. The authors have definitely done justice in covering several transcriptional factors and epigenetic mechanistic pathways in their review. However, the English language requires improvement in several sections. I suggest you have a colleague who is proficient in English and familiar with the subject
matter review your manuscript, or contact a professional editing service.

Few examples,
a. Abstract line 15 and line 23 have grammatical errors.
b. Introduction line 35 " Epigenetics is a..." It is a very long sentence with grammatical errors. Line 41 last part is a repetition of abstract last line.
c. Survey methodology line 45 should be 'contains' instead of 'contained'.
d. Line 95- There should be 'the'
e. Line 116 it should be dimerization and not dipolymerization
f. Line 121 Moreover, it was observed ...is incomplete
g. Line 147 there is no word called specificality
h. Line 169- grammatical error in sentence
i. Line 209- 'Over 90% of..' grammatical error present
j. Line 233- 'It reported' is incorrect

Authors must improve the flow of English language of the review.

2. Certain concepts mentioned in the review are either incorrect or contradictory

Few examples:
1. Line 53 astrocytes have many neurites is incorrect
2. Line 125 NSC is introduced here but there is no full form
3. Line 143 how does UAC1 overexpression inhibit astrocyte activation via JAK-STAT pathway. There needs to be more elaboration on this concept
4. in vitro and in vivo should always be italized
5. SVZ has not been expanded
6. Line 175 there is contradiction in the statements. Ngn2 is a repressor of astrogliogenesis, however the sentence says mediates gliogenesis.
7. Line 188 - Again contradiction. Is there upregulation or downregulation of astrogliogenesis with the deletion of Zbtb20?
8. Line 189- ZBTB20 gene caused Primrose syndrome which astrogliogenesis is
190 severely disrupt. How is it disrupted? Elaborate
9. Figure 3 legend says- 'Knockout of DNMT1, DNMT3a, and DNMT3b make demethylation on GFAP promoter and promote the transcription of GFAP. The activation of STAT3 promotes the hypomethylation of GFAP promoter'. Line 212 says 'Knockout DNMT1, DNMT3a and DNMT3b in mouse embryonic stem cells detected complete demethylation on the GFAP promoter region, thus, STAT3 failed to bind to the GFAP promoter to induce GFAP transcription'. The figure and the sentences are contradictory.

Experimental design

Rigorous literature review has been conducted and the content is within the Aims and scope of the journal

Validity of the findings

The review article is useful for future research on glial therapies.

The article fails to clearly state the facts. The flow of the review article is not clear in many sections.

Reviewer 3 ·

Basic reporting

In this review, Ma et al. discussed transcriptional and epigenetic regulation pathways in astrocyte function. While the topic is interesting and novel, the paper contains much text just listing results from different studies without logical organization. Additionally, the English language should be improved throughout the manuscripts, for which requires input from a colleague who is proficient in English or contact a professional editing service. Collectively, the authors should refine their logic and emphasize the unique contributions of their review.

Experimental design

Although the authors claimed that ‘This review contained the research articles from the past five years’, some recent key findings were not discussed. E.g. PMID: 37319217.

Validity of the findings

N/A

Additional comments

Introduction: To enhance the clarity and relevance of the introduction, the authors should articulate early on what novel insights this review brings to the field of astrocyte research. A substantial portion of the introduction was devoted to listing findings of various literature without clear logic and the exact rationale for why this topic is important to the field. Another minor comment: the structure of the introduction seems unconventional with three sections (Rationale, intended audience, and survey methodology).

Abstract: The abstract fails to capture the novelty and significance of this review and requires language edits.

Main text just lists results from different studies without logical organization and connections between each section. Some key or novel factors for astrocyte transcriptional pathways (e.g. Sox9) and epigenetic pathways (e.g. serotonylation) were not discussed..

Figures: It would be helpful for readers to have figures illustrating transcriptional/epigenetic pathways discussed in the review.

·

Basic reporting

The manuscript by Ma et al provides an interesting oveerview of the subject of astrogliogenesis and epigenetics, an interesting field that has not been reviewed lately. The authors do a good job summarizing all the literature available and the figures are explanatory and complement the manuscript . However, major changes on redaction and synthaxis are needed;
On introduction
Major reviews:
Line 53: Astrocytes have many neurites. Neurites can only exist within the context of a neural cell. Change the term to projections.


Minor changes:
Line 35 "Neuroscience is an active research field with many fascinating yet unresolved issuesÿ Epigenetics is a". The comma should be a point

Line 35: Rephrase or divide this sentence in 2 "Epigenetics is a
discipline that has gradually developed in recent years, developing through the study of many life phenomena that do not conform to the classical Mendelian genetic laws including DNA, RNA interference, and histone modifications, play the important role and make the huge progress in neurosciences including astrogliogenesis, exhibit the enormous application potential in the study of astrogenesis mechanisms"

Line 41: Rewrite "hope the review may provide some useful clue for the future
42 research". Though true, find a better way to phrase this sentiment.


Line 47: "Astrocytes are the most widely distributed and have the largest size among glial cell types", largest is already a measurement of size, propose to say are the most widely distributed and largest among...

Experimental design

Major reviews:

At line 208 "At E11.5 days, the GFAP promoter bound by STAT3 was rapidly demethylated in response to STAT3 signaling activation" please state who did the experiment and the model (assuming mouse), but seems to be based only on one paper. Provide more context

Minor reviews:
Avoid using of contractures (can't) etc. Examples on line 69,

Line 97 "Studies above suggest NFIA is a so crucial and indispensable transcription factor during the process o" remove the so before crucial.


At line 233 "It reported that H3K27ac and H3K4me1 exist crosstalk during the onset of astroglial differentiation, making us understanding the transcription of astroglial genes post epigenetic chromatin marks" the meaning of this phrase is unclear. Please rewrite.

line 245: "Studies have shown that MAPK signaling activated by FGF in cortical development, which is critical to" please change to "Studies have shown that MAPK signaling is activated by FGF in cortical development, which is critical to"

line 260 :MiR-153 also targeted NeuroD1 in glial cells was involved in LPS-induced neuroinflammatory through inhibiting th" Be consistent with miR153 , use a lowercase even after a point. Also add a preposition (and) between cells and was involved in..

Line 263-265 : the phrase is too dense. Hard to read, divide it into 2 or 3 different ones.

Line 275 change to "These results implicate that miR-124 facilitates neuronal differentiation by targeting"

Validity of the findings

Conclusions require a major write up and are not measuring up to the great work the authors did by summarizing all the previous complex information. Especially the first lines.

---

## Round 0.2 · accepted · Accept

· Academic Editor

Accept

Dear Dr Yang,

Thank you for submitting your revised manuscript "The role of transcriptional and epigenetic modifications in astrogliogenesis". It has now been seen by two of the original referees and their comments are below. The reviewers find that the paper has improved in revision, and therefore we are happy in principle to publish it in PeerJ.

Thank you again for your interest in PeerJ. Please do not hesitate to contact me if you have any questions.

Sincerely,

Alexis Verger

·

Basic reporting

Overall great improvement.

Abstract line 23 has grammatical error. Sentence' hope it enhances..' needs to be rephrased

Experimental design

All good

Validity of the findings

all good

Reviewer 3 ·

Basic reporting

The revised manuscript has been significantly improved. This review covers an interesting topic and would be informative for a wide range of readers.

Experimental design

The authors have updated the literature search to address reviewer's comments.

Validity of the findings

n/a